# Pathological Relationship between Intracellular Superoxide Metabolism and p53 Signaling in Mice

**DOI:** 10.3390/ijms22073548

**Published:** 2021-03-29

**Authors:** Kenji Watanabe, Shuichi Shibuya, Yusuke Ozawa, Toshihiko Toda, Takahiko Shimizu

**Affiliations:** 1Aging Stress Response Research Project Team, National Center for Geriatrics and Gerontology, Obu 474-8511, Aichi, Japan; kng-wtnb@ncgg.go.jp (K.W.); s-shibuya@ncgg.go.jp (S.S.); 2Department of Endocrinology, Hematology and Gerontology, Chiba University Graduate School of Medicine, Chiba 260-8677, Chiba, Japan; ozawayusuke3@gmail.com (Y.O.); hik_toda@proteome.jp (T.T.)

**Keywords:** superoxide dismutase (SOD), p53, superoxide, aging, apoptosis

## Abstract

Intracellular superoxide dismutases (SODs) maintain tissue homeostasis via superoxide metabolism. We previously reported that intracellular reactive oxygen species (ROS), including superoxide accumulation caused by cytoplasmic SOD (SOD1) or mitochondrial SOD (SOD2) insufficiency, induced p53 activation in cells. SOD1 loss also induced several age-related pathological changes associated with increased oxidative molecules in mice. To evaluate the contribution of p53 activation for SOD1 knockout (KO) (*Sod1^−^*^/*−*^) mice, we generated SOD1 and p53 KO (double-knockout (DKO)) mice. DKO fibroblasts showed increased cell viability with decreased apoptosis compared with *Sod1^−^*^/*−*^ fibroblasts. In vivo experiments revealed that p53 insufficiency was not a great contributor to aging-like tissue changes but accelerated tumorigenesis in *Sod1^−^*^/*−*^ mice. Furthermore, p53 loss failed to improve dilated cardiomyopathy or the survival in heart-specific SOD2 conditional KO mice. These data indicated that p53 regulated ROS-mediated apoptotic cell death and tumorigenesis but not ROS-mediated tissue degeneration in SOD-deficient models.

## 1. Introduction

Age-related pathological changes are caused by several genetic and environmental factors. To analyze the age-related changes in vivo and in vitro, researchers have used several genetic and pharmacological manipulations for the induction of redox imbalance [1,2]. Superoxide dismutase (SOD) enzymes play a major role in the intracellular antioxidant system by catalyzing the conversion of superoxide radicals (O_2_^•−^) to hydrogen peroxide and O_2_ [3]. In mammals, copper/zinc-SOD (SOD1) exists in the cytoplasm, while manganese-SOD (SOD2) is distributed in the mitochondrial matrix to regulate intracellular redox balance in cells. Since SOD expression and activity are significantly decreased in aged osteoporotic, end-stage osteoarthritic, and Alzheimer’s disease individuals [4,5,6], redox imbalance caused by SOD decline is considered an important mechanism underlying the induction of age-related pathological changes.

We previously reported that SOD1-deficient mice showed the accumulation of oxidative molecules and several age-related pathological changes, including macular degeneration [7,8], hemolytic anemia with splenomegaly [9], osteopenia [10,11], skin atrophy [12], skeletal muscle atrophy [13], hepatic carcinoma [14], and fatty liver [15]. In addition, SOD1-deficient phenotypes can be improved by antioxidant treatments in vivo [12,16,17,18,19]. Therefore, *Sod1^−^*^/*−*^ mice are a useful model for age-related tissue deregulation and intervention strategies. In addition, we previously reported that *Sod1^−^*^/*−*^ fibroblasts showed the significant presence of intracellular reactive oxygen species (ROS), including O_2_^•−^ accumulation accompanied by p53 upregulation, which resulted in apoptotic cell death [20]. A rescue experiment using antioxidant reagents exhibited effective suppression of p53 activation and cell death in *Sod1^−^*^/*−*^ fibroblasts [20]. These data suggest that p53 plays a fundamental role in SOD1 loss-related phenotypes. However, whether or not p53 activation directly regulates SOD1-deficient phenotypes in vivo and in vitro remains unclear.

In contrast to the above findings, SOD2-deficient mice showed dilated cardiomyopathy (DCM), steatosis, and metabolic acidosis, which resulted in neonatal lethality [21]. To analyze the SOD2-deficient phenotypes in adults, we generated and established tissue-specific SOD2 knockout (KO) mice [2,22]. Several tissue-specific SOD2-deficient mice showed DCM-type heart failure [23], disturbance of exercise activity [24], spongiform encephalopathy [25], bone loss [26], and cartilage degeneration [27]. Consequently, we proposed that tissue-specific SOD2 KO mice were a useful model of age-related pathological changes caused by mitochondrial dysfunction. In addition, pressure overload, hypoxic stress, and genotoxic stress-induced p53 upregulation resulted in cardiomyocyte death [28]. Heart failure such as DCM and right ventricular hypertrophy model mice also showed the accumulation of p53 and cardiomyocyte apoptosis [29,30,31,32]. Since suppression of p53 via pharmacological as well as genetic approaches ameliorated heart failure [29,33,34], p53 is considered a key molecule involved in heart failure. However, while SOD2 loss induced DCM in mice, the involvement of p53 in DCM-type heart failure caused by SOD2 loss remains unclear.

In the present study, in order to clarify the contribution of p53 to the *Sod1*- or *Sod2*-deficient phenotypes in vivo, we generated two types of double-knockout (DKO) mice: *Sod1* and *p53* DKO mice, as well as heart-specific *Sod2* and *p53*-deficient mice. We also discussed the influence of p53 deficiency on the phenotypes of SOD1 or heart-specific SOD2 KO mice.

## 2. Results

### 2.1. p53 Insufficiency Effectively Suppressed Apoptotic Cell Death In Vitro

Previously, we reported that the intrinsic O_2_^•−^ accumulation by SOD1 loss promoted p53 activation and apoptotic cell death in vitro [20]. In addition, antioxidant reagents effectively attenuated *Sod1*-deficient phenotypes accompanied by p53 upregulation in fibroblasts and skin tissues [20]. To clarify the pathological relationship of p53 upregulation in Sod1-deficient phenotypes in mice, we generated *Sod1* and *p53* DKO mice. First, we performed in vitro fertilization of *Sod1* KO oocytes with *p53* KO frozen sperm to obtain double-heterozygous mice. Next, we intercrossed double-heterozygous males and females to generate DKO mice. Unexpectedly, we obtained only a very small number of DKO mice from cross-fertilization via natural mating as well as in vitro fertilization (Table 1). These data indicated that the birth rate of DKO mice was not Mendelian.

Next, we assessed the efficiency of p53 loss in *Sod1^−^*^/*−*^ fibroblasts. Although *Sod1^−^*^/*−*^ fibroblasts died within 3 days under 20% O_2_ conditions, p53 loss improved the cell number decline among *Sod1^−^*^/*−*^ cells (Figure 1A). However, DKO cells showed low cell proliferation as well as significant incrementation of dihydroethidium (DHE)- and CM-H_2_DCFDA (DCF)-positive ROS, including O_2_^•−^ accumulation, with values similar to those seen in *Sod1^−^*^/*−*^ cells (Figure 1B–D). Interestingly, DKO cells showed significantly fewer apoptotic cells than *Sod1^−^*^/*−*^ cells (Figure 1E). In contrast, *p53^−^*^/*−*^ cells exhibited no harmful phenotypic effect, including with regard to the cell proliferation and ROS accumulation (Figure 1B–D). These data indicated that p53 impairment suppressed apoptotic cell death, which resulted in an increase in the cell survival among DKO cells.

### 2.2. p53 Loss Failed to Attenuate the SOD1-Deficient Phenotypes in Mice

To evaluate the effect of p53 deficiency in *Sod1^−^*^/*−*^ mice, we expanded the intercrossing and analyzed organ phenotypes of DKO mice. *Sod1^−^*^/*−*^ mice revealed body weight reduction, muscle atrophy, and liver weight gain [1], but these were not significant differences compared with wild-type (WT) mice in this analysis (Figure 2A–C). SOD1 loss significantly induced skin thinning and decrease of red blood cell number but not splenomegaly (Figure 2D–F). On the other hand, *p53^−^*^/*−*^ mice showed no significant differences in all parameters (Figure 2A–F). DKO mice showed significant reductions of muscle weight, skin thickness, and red blood cell number (Figure 2B,D,F). Interestingly, DKO mice also exhibited exacerbation of splenomegaly compared with *Sod1^−^*^/*−*^ mice (Figure 2E). Importantly, p53 haploinsufficiency also failed to improve *Sod1^−^*^/*−*^ phenotypes (Figure 2A–F). Furthermore, *Sod1^+^*^/*−*^*, p53^−^*^/*−*^ mice were extremely similar to WT and *p53^−^*^/*−*^ mice (Figure 2A–F). These data indicate that p53 insufficiency did not seriously influence the organ phenotypes of *Sod1^−^*^/*−*^ mice.

### 2.3. SOD1 and p53 DKO Mice Showed Early Tumor Progression

About half of p53 KO mice reportedly show tumor progression by six months of age [35]. In contrast, *Sod1* KO mice have been reported to reveal no tumor phenotypes until six months of age [14]. We therefore monitored the tumor progression phenotypes in DKO mice until four months of age. A large number of DKO mice showed remarkable spontaneous tumor progression in the appearance of discriminative by four months of age (Table 2 and Figure 3A). Whereas p53 KO mice mostly showed thymic lymphoma or sarcomas [35], DKO mice developed multifarious tumor throughout the whole body, including in the cervix, abdomen, limbs, and testis (Figure 3A,B). Importantly, *Sod1^−^*^/*−*^, *p53^+^*^/*−*^ as well as *Sod1^+^*^/*−*^*, p53^−^*^/*−*^ mice displayed no tumor progression by four months of age (Table 2), suggesting that heterozygotic loss of *p53* or *Sod1* was sufficient to achieve the suppression of tumor development in DKO mice. These data indicated that systemic oxidative damage caused by complete SOD1 loss accelerated the tumor initiation and/or development in the whole body of *p53^−^*^/*−*^ mice.

### 2.4. p53 Insufficiency Had No Effect on the Heart Failure of Heart-Specific Sod2-Deficient Mice

We previously found that heart-specific *Sod2*-deficient (*Sod2^H^*^/*H*^) mice showed a short lifespan associated with DCM [23]. Accumulating evidence has suggested that heart failure involves the p53 signaling pathway [28]. In vitro studies revealed that *Sod2* loss increased mitochondrial ROS and p53 activation in mouse embryonic fibroblasts (Watanabe et al., personal communication). In this context, to clarify the contribution of p53 to heart failure in *Sod2^H^*^/*H*^ mice, we generated heart-specific *Sod2*- and *p53*-deficient mice (*Sod2^H^*^/*H*^*, p53^H^*^/*H*^). *Sod2^H^*^/*H*^*, p53^H^*^/*H*^ mice had a similarly short lifespan to *Sod2^H^*^/*H*^ mice (Figure 4A). Furthermore, DCM caused by heart-specific *Sod2* loss was also recognized in *Sod2^H^*^/*H*^*, p53^H^*^/*H*^ mice (Figure 4B,C). *p53^H^*^/*H*^ mice showed a normal lifespan and heart tissue structures with strong similarity from those of WT mice including *Sod2^f^*^/*f*^ and *p53^f^*^/*f*^ mice (Figure 4A–C). Importantly, the pathogenesis of cardiac fibrosis was also not markedly different between *Sod2 ^H^*^/*H*^ and *Sod2^H^*^/*H*^*, p53^H^*^/*H*^ mice (Figure 4C). These data indicated that the induction and the progression of DCM phenotypes by *Sod2* in mice were not influenced by the loss of the p53 molecule.

## 3. Discussion

It is very well known that p53 is involved in several signaling pathways, including the DNA damage response (DDR) leading to cellular senescence induction, cell cycle arrest, DNA repair, autophagy, and cell death [36,37]. Previously, we reported that SOD1 loss induced the marked intracellular accumulation of ROS (about 40-fold) accompanied by p53 activation in vitro [1,20], suggesting a close relationship between the induction of SOD1-decifient phenotypes and p53 activation in vitro. p53 regulates the cell fate, such as the transcriptional induction of antioxidant-, cell cycle arrest-, and apoptosis-related genes, according to the intracellular redox state [34,38]. Low levels of ROS accumulation induced p53-mediated cytoprotective property and suppressed apoptosis [39]. In addition, moderate ROS activated cell cycle checkpoint genes, which resulted in cell cycle arrest for DNA repair [40]. In contrast, excessive ROS stress was shown to lead to apoptosis [39]. Whereas *p53* loss remarkably increased the survival of *Sod1^−^*^/*−*^ fibroblasts to a point similar to that of WT cells, *p53* insufficiency did not influence the intracellular ROS accumulation or cell proliferation caused by SOD1 loss (Figure 1). p53 deficiency is well known to be incapable of promoting apoptotic cell death caused by ROS in fibroblasts [41,42]. Our data also indicated that *p53* deficiency effectively suppressed apoptosis induction via DDR in *Sod1^−^*^/*−*^ cells. Therefore, DKO cells can survive despite oxidative damage under normal atmospheric conditions (Figure 1). These results indicated that p53 mainly regulated apoptotic cell death rather than cell cycle arrest when excessive intracellular ROS accumulated in *Sod1^−^*^/*−*^ cells.

However, an in vivo study showed that p53 did not strongly influence SOD1-deficient phenotypes (Figure 2). Since *p53* deficiency influenced erythrocytes turnover and eryptosis induction [43], DKO mice might show an increased rate of splenomegaly accompanied by a tendency toward decreased red blood cell numbers (Figure 1D,E). Actually, *Sod1^−^*^/*−*^ mice showed about 1.5 times the levels of systemic oxidative markers (8-isoprastane, malondialdehyde, 4-hydroxyalkenals, and 8-hydroxy-2′-deoxyguanosine (8-OHdG)) compared with WT mice [16,18,44,45]. In addition, SOD1-deficient tissues, such as the lacrima gland, liver, skin, and ovary, also showed relative low levels of oxidative damage [15,16,19,45,46], indicating that intravital oxidative stress was moderate compared with cellular oxidative stress in vitro. Because the intravital O_2_ concentration remained low [47], the extent of oxidative damage caused by SOD1 loss might be too small to induce DDR-mediated apoptosis in mice. Previously, we reported that SOD1 loss activated the Forkhead box O3 (FoxO3)-metalloproteinase-2 (Mmp2) axis, resulting in skin thinness [17]. This indicated that the transcriptional factor FOXO3a and not p53 regulated skin atrophy caused by SOD1 loss.

Since SOD1 enzyme includes copper and zinc ions, SOD1 also acts as a chelator of copper and zinc ions. *Sod1* deficiency might induce an increase in free copper and zinc ions in cytoplasm. Overdose of copper induced apoptotic cell death in granule cells, resulting in degeneration and neuronal loss in the central nervous system [48]. Furthermore, excessive zinc induced the disturbance of redox balance, gene expression, bone metabolism, and alternation of the p53 protein structure [49,50,51]. Several age-related chronic diseases also showed increased serum levels of copper and zinc ions [52,53]. In this context, ion homeostasis failure caused by SOD1 loss might induce age-related pathological changes in vivo and in vitro. Recently, several studies reported that SOD1 protein induces post-translational modifications and regulates the expression of antioxidant genes as a transcriptional factor [54]. In addition to the loss of antioxidant activity, the loss of the metal chelating ability and transcriptional function might markedly affect *Sod1*-deficient phenotypes. Further studies are needed to clarify the molecular mechanisms involved in *Sod1*-defcient phenotypes.

In contrast to the above findings, our results showed that *Sod1* deficiency exacerbated tumor progression in *p53^−^*^/*−*^ mice (Table 2). A high percentage (68%) of *Sod1^−^*^/*−*^ mice revealed nodular hyperplasia or hepatocellular carcinoma by 20 months of age [14], whereas DKO mice showed the early detection of tumor formation (by 4 months) with a high probability (50%) (Table 2 and Figure 3). Because p53 protects tumorigenesis from ROS-mediated DNA damage [55], the significant increase in oxidative damage induced by SOD1 insufficiency may accelerate tumorigenesis in *p53^−^*^/*−*^ mice. Furthermore, E2-promoter binding factor (E2F) transcriptional factor interacted with retinoblastoma susceptibility genes to regulate cellular proliferation and tumorigenesis [56,57]. Double mutant for the E2F family of transcription factors, including E2F1 and E2F2, resulted in γH2AX accumulation accompanied by p53 activation, which consequently caused apoptotic cell death in the pancreas [58]. Disruption of p53 in E2F1 and E2F2 double-knockout mice caused the suppression of apoptosis induction, which resulted in the progression of thymic lymphomas and a shortened lifespan [58]. This suggests that p53-dependent apoptosis induced by SOD1 or E2F1/E2F2 deficiencies is a key mechanism underlying tumor suppression. Accordingly, p66Shc generates hydrogen peroxide, and p66Shc loss decreases ROS production. In this context, p66Shc insufficiency significantly increased the lifespan and suppressed tumor progression in *p53* KO mice [59].

Many reports have shown that heart failure as age- or pathology-related phenotypes is mediated by p53 upregulation [28]. In addition, p53 suppression by pharmacological and genetic techniques ameliorates the phenotypes in heart failure models [28]. In this context, we generated *Sod2^H^*^/*H*^*, p53^H^*^/*H*^ mice to attenuate the DCM phenotypes of *Sod2^H^*^/*H*^ mice. Unexpectedly, p53 loss failed to improve the short lifespan and DCM phenotypes in *Sod2^H^*^/*H*^ mice (Figure 4). In general, hypoxic and genotoxic stress induces cardiomyocyte apoptosis through p53 activation, resulting in heart failure [29,30]. However, since *Sod2^H^*^/*H*^ mice did not show the induction of apoptotic cell death in the heart [23], p53 insufficiency may not mitigate heart failure in *Sod2^H^*^/*H*^*, p53^H^*^/*H*^ mice. We previously reported that antioxidant reagents, such as EUK-8, MnTBAP, manganese porphyrins, and apple procyanidins, improved DCM accompanied by a reduction in mitochondrial ROS accumulation in *Sod2^H^*^/*H*^ mice [23,60,61,62]. Recently, Guo et al. reported that a loss-of-function mutation in extracellular SOD (SOD3) induced chronic kidney disease accompanied by systolic hypertension and cardiac hypertrophy in a Dahl/salt-sensitive strain of rats [63]. In addition, SOD3 KO mice also showed hypoxia-induced pulmonary vascular disease [64,65]. These reports suggest that not only intracellular but also extracellular ROS affect cardiac hypertrophy and cardiovascular diseases.

Taken together, our data indicate that p53 plays a minimal role in the pathogenesis of SOD1 or heart-specific SOD2 deficiency in mice. Since p53 mainly functions in apoptosis induction and tumor suppression, it has little involvement in apoptosis-independent tissue disorders, including adult *Sod1^−^*^/*−*^ mice and *Sod2^H^*^/*H*^ heart. In contrast, the suppression of apoptosis by p53 loss accelerated tumor initiation/progression in *Sod1^−^*^/*−*^ mice. Likewise, SOD1 deficiency accelerated tumor progression in *p53^−^*^/*−*^ mice, indicating that apoptosis induction by p53 as well as intracellular O_2_^•−^ metabolism by SOD1 strongly contributed to tumor suppression. In conclusion, SOD1-deficient mice and tissue-specific SOD2-deficiet mice were useful model mice for an aging study without tumor progression.

## 4. Materials and Methods

### 4.1. Animals

*Sod1^−^*^/*−*^ mice were generated by intercrossing *Sod1^+^*^/*−*^ males and females (Jackson Laboratory, Bar Harbor, ME, USA). We mated *Sod1^−^*^/*−*^ mice with C57BL-p53+/− mice (RBRC01361, RIKEN BRC, Ibaraki, Japan) [35] to generate the *Sod1* and *p53* DKO mice. We crossed heart-specific *Sod2-*deficient mice (*Sod2^H^*^/*H*^) [23] with *Trp53* flox mice (Stock no. 008462, Jackson Laboratory) [66] to obtain heart-specific *Sod2* and *p53* DKO mice (*Sod2^H^*^/*H*^*, p53^H^*^/*H*^). All of the genotypes of *Sod1*, *p53*, Cre recombination transgene, *Sod2* flox, and *p53* flox mice were assessed by polymerase chain reaction (PCR) using genomic DNA isolated from the tail tip, as described previously [11,23,35,66]. Primer sequences are given in Appendix A. The animals were housed under a 12-h light/dark cycle and fed *ad libitum*. The experimental procedures were approved by the Animal Care and Use Committee of Chiba University and National Center for Geriatrics and Gerontology.

### 4.2. Histology

For histological morphology, skin specimens from the back tissue, heart, and tumor were dissected and fixed in a 20% formalin neutral buffer solution (FUJIFILM Wako, Osaka, Japan) overnight while embedded in paraffin and then sectioned on a microtome at 4 µm according to standard techniques. Hematoxylin and eosin staining for the skin morphology and heart as well as Azan staining for total collagen deposition were performed as described previously [15,23,67]. The thickness of the skin tissue was determined using the BZ-X Analyzer software program (Keyence, Osaka, Japan).

### 4.3. Cell Culture

The skin tissue specimens were dissected from 5-day-old neonates. The primary dermal fibroblasts were isolated by dissociation in 0.2% collagenase type 2 (Worthington Biochemical Corporation, Lakewood, NJ, USA) at 37 °C for 60 min. The cells were cultured in minimum essential medium Eagle, alpha modification (α-MEM; Life Technologies Corporation, Carlsbad, CA, USA) supplemented with 20% fetal bovine serum (FBS, Thermo Fisher Scientific, Waltham, MA, USA), 100 unit/mL penicillin, and 0.1 mg/mL streptomycin at 37 °C in a humidified incubator (ASTEC, Fukuoka, Japan) with 5% CO_2_ and 1% O_2_ to expand and maintain *Sod1^−^*^/*−*^ fibroblasts. During experiments, the cells were cultured under 20% O_2_ conditions. Cell viability was measured by the cell proliferation enzyme-linked immunosorbent assay bromodeoxyuridine (BrdU; Roche Diagnostics K.K., Basel, Switzerland) according to the manufacturer’s instructions. The relative BrdU incorporate values were calculated by a triplicate analysis.

### 4.4. Flow Cytometry

The accumulation of intracellular ROS was detected using DHE and DCF (Life Technologies Corporation). The cells were incubated with 10 μM DHE or 10 μM DCF for 30 min at 37 °C. Following incubation, the cells were trypsinized and resuspended in phosphate-buffered saline. Apoptosis was measured using a fluorescein isothiocyanate (FITC) Annexin V Apoptosis Detection Kit I (BD Biosciences, San Diego, CA, USA) according to the manufacturer’s instructions. The fluorescence intensities were assessed using a flow cytometer (BD FACSCanto II; BD Biosciences, San Diego, CA, USA).

### 4.5. Statistical Analyses

Statistical evaluations were performed using a two-way analysis of variance with the GraphPad Prism9 software program (GraphPad Software, San Diego, CA, USA). Differences between the data were considered to be significant when the *p*-values were less than 0.05. The data are represented as the means ± the standard deviation (SD).

## Figures and Tables

**Figure 1 ijms-22-03548-f001:**
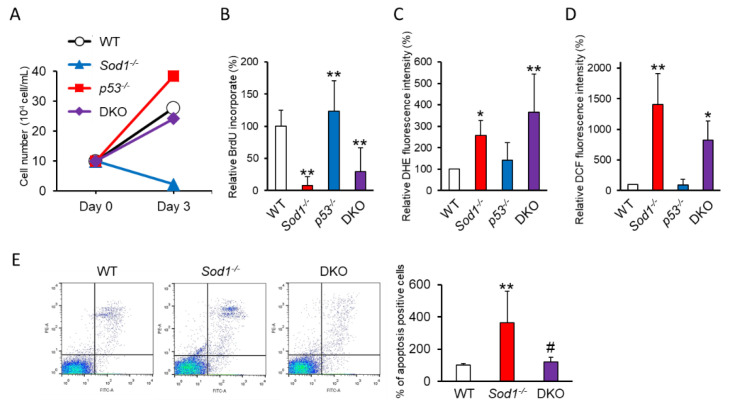
p53 insufficiency suppressed cytoplasmic superoxide dismutase (SOD1) loss-mediated apoptotic cell death. (**A**) The cell numbers of wild-type (WT); *Sod1^−^*^/*−*^*, p53^−^*^/*−*^; and double-knockout (DKO) fibroblasts were counted at the times indicated. (**B**) The cell viabilities of each type of fibroblast were measured based on the incorporation of bromodeoxyuridine (BrdU) into the fibroblasts. (**C**,**D**) The intracellular superoxide accumulation was measured by flow cytometry with dihydroethidium (DHE) (**C**) and CM-H_2_DCFDA (DCF) (**D**) in each type of fibroblast. (**E**) Apoptosis in fibroblasts was analyzed by flow cytometry with propidium iodide (PI) and annexin V. Statistical analyses were performed using a two-way analysis of variance. The error bars indicate the standard deviation of three independent experiments (n = 3). * *p* < 0.05, and ** *p* < 0.01 vs. WT. # *p* < 0.05 vs. *Sod1^−^*^/*−*^.

**Figure 2 ijms-22-03548-f002:**
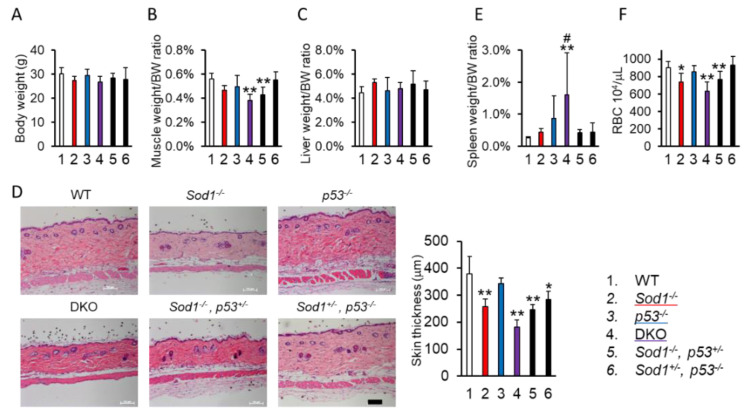
p53 loss had no marked effect on SOD1-decifient phenotypes in mice. (**A**) The body weight of each mouse (at four months of age). The genotypes for each mouse were as follows: (1) Wild-type (*n* = 10); (2) *Sod1^−^*^/*−*^ (*n* = 7); (3) *p53^−^*^/*−*^ (*n* = 5); (4) *Sod1^−^*^/*−*^*, p53^−^*^/*−*^ (DKO, *n* = 5); (5) *Sod1^−^*^/*−*^*, p53^+^*^/*−*^ (*n* = 10); (6) *Sod1^+^*^/*−*^*, p53^−^*^/*−*^ (*n* = 9). (**B**) The ratio of muscle weight corrected by body weight. (**C**) The ratio of liver weight corrected by body weight. (**D**) Hematoxylin and eosin staining, the thickness of the back skin, and the skin thickness of each mouse. (**E**) The ratio of the spleen weight corrected by the body weight. (**F**) The number of red blood cells in each mouse. Statistical analyses were performed using a two-way analysis of variance. The error bars indicate the standard deviation. * *p* < 0.05. ** *p* < 0.01 vs. WT. ^#^
*p* < 0.05 vs. *Sod1^−^*^/*−*^. The scale bar represents 100 μm.

**Figure 3 ijms-22-03548-f003:**
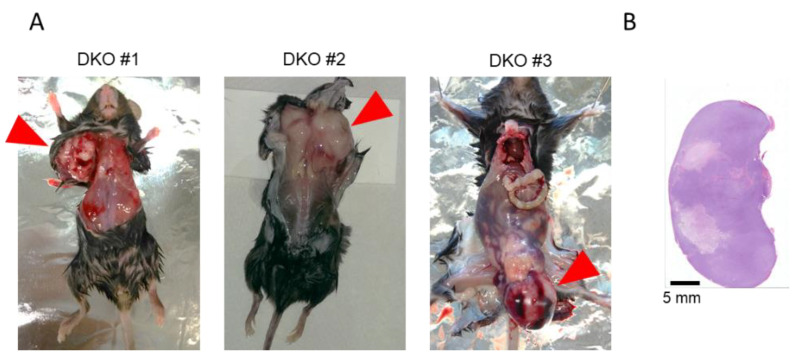
SOD1 loss accelerated tumor progression in p53 KO mice. (**A**) Representative DKO mice accompanied by a tumor. The red arrows indicate the tumor. (**B**) Hematoxylin and eosin staining of the tumor from DKO mice.

**Figure 4 ijms-22-03548-f004:**
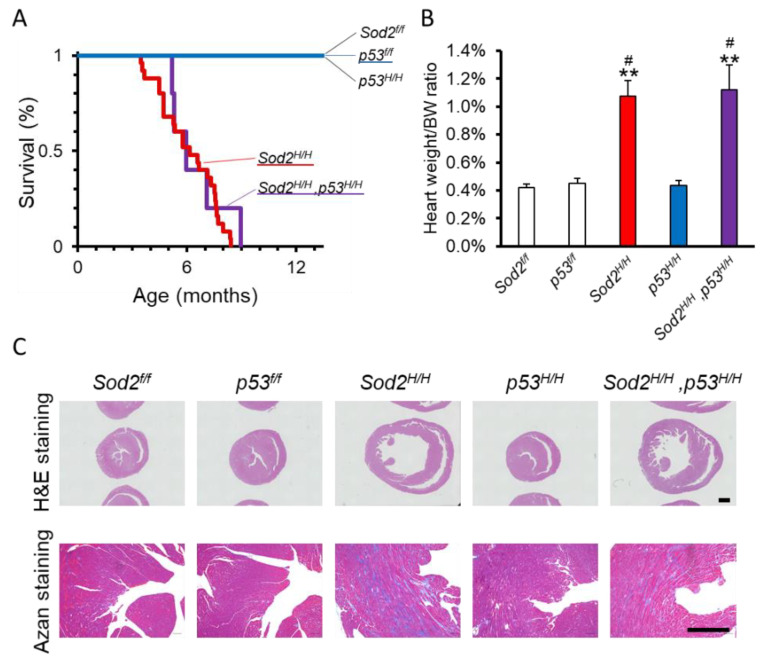
The mitochondrial SOD (SOD2)-deficient phenotypes were largely unaffected by p53 loss. (**A**) A comparison of the survival curves in each mouse. A Kaplan–Meier analysis was used to estimate the median lifespan. The numbers of individuals in each group were as follows: *Sod2^f^*^/*f*^ (*n* = 8); *p53^f^*^/*f*^ (*n* = 8); *Sod2^H^*^/*H*^ (*n* = 7); *p53 ^H^*^/*H*^ (*n* = 6); *Sod2 ^H^*^/*H*^*, p53 ^H^*^/*H*^ (*n* = 11). (**B**) The ratio of heart weight corrected by body weight. (**C**) Hematoxylin and eosin staining as well as Azan staining of the heart of each mouse. Statistical analyses were performed using a two-way analysis of variance. The error bars indicate the standard deviation. ** *p* < 0.05 vs. *Sod2^f^*^/*f*^, and # *p* < 0.05 vs. *p53^f^*^/*f*^. The scale bars represent 1 mm ((**C**), upper panel) and 500 μm ((**C**), lower panel).

**Table 1 ijms-22-03548-t001:** The genotype of births to the cross-breeding between double-heterozygous mice.

		*Sod1*		
		+/+	+/−	−/−
*p53*	+/+	12	21	13
	+/−	27	44	20
	−/−	7	10	2

**Table 2 ijms-22-03548-t002:** The number of tumor or death to each mouse until 4 months of age.

Genotype	Number	Tumor	Death
WT	10	0	0
*Sod1^−^* ^/*−*^	7	0	0
*p53^−^* ^/*−*^	6	1 (17%)	0
DKO	14	7 (50%)	2 (14%)
*Sod1^−^* ^/*−*^ *, p53^+^* ^/*−*^	10	0	0
*Sod1^+^* ^/*−*^ *, p53^−^* ^/*−*^	9	0	0

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
