# Peer review of "Pathological Relationship between Intracellular Superoxide Metabolism and p53 Signaling in Mice"

_ijms, 2021, doi:10.3390/ijms22073548_

Round 1

Reviewer 1 Report

The manuscript by Watanabe et al investigates on the possible interaction between p53 and the functions supported by SOD1 or SOD2. To this aim, mice were generated having a double knock out for SOD1 and p53, or for SOD2 and p53. However, as stated by the same Authors, the in vivoand in vitroresults discussed in the manuscript suggest that p53 only plays a minimal role in the pathogenesis related to SOD1 or SOD2 deficiency. Therefore, this finding isn’t in good harmony with the title given to the article, because no evident cross-talk exists between p53 and SOD metabolism.

However, the manuscript is interesting because faced with a proper strategy and therefore could merit publication on Int. J. Mol. Sci.if the following concerns are carefully addressed.

-   Modify the title of the manuscript with a less ambitious sentence.

-   Introduction. The Authors did not discuss the properties of another important SOD, such as the extracellular enzyme (SOD3) exerting a key role in the redox control of extracellular space. In particular, it is known that SOD3 deficiency is involved in several pathologies, including cardiovascular diseases (Eur. J. Histochem 2014, 58:2383; Sci. Rep. 2020 10, 210). Therefore, this item, although not considered in the experimental strategy, should be at least discussed.

-   Fig. 1. In panel B the statistical significance (p < 0.01) assigned to p53–/–isn’t believable with the presented error bars.

-   Fig. 1. In panel E, an obvious positive control (p53–/–) is missing; please, add this information or explain the reasons for its omission.

-   Results section (page 3, lines 100-102). The statement “Unexpectedly, p53 loss did not induce SOD1-deficient phenotypes, such as body weight reduction, muscle atrophy, and liver weight gain, …” is unclear. Indeed, the data presented in Fig. 2A-C only show a minimum and not significant difference between histograms of WT and Sod1–/–; therefore, the comment related to the missing induction due to the loss of p53 (histograms of p53–/–) seems not appropriate. Only, the comment related to the skin thickness seems to be appropriate.

-   Fig. 2. Please revise the indication for panels E and F, because there is an apparent inversion of letters in Figure legend.

-   Results section (page 5, paragraph 2.4). Please, add the meaning for genotypes p53f/fand Sod2f/fin the text.

-   Fig. 4. In panel A, lines related to Sod2f/fand p53f/fare missing.

-   Discussion. This part of the manuscript seems extremely long. This Reviewer suggests a significant shortage to allow a better focusing on the objectives reached with the research.

Author Response

Reviewer 1

The manuscript by Watanabe et al investigates on the possible interaction between p53 and the functions supported by SOD1 or SOD2. To this aim, mice were generated having a double knock out for SOD1 and p53, or for SOD2 and p53. However, as stated by the same Authors, the in vivoand in vitroresults discussed in the manuscript suggest that p53 only plays a minimal role in the pathogenesis related to SOD1 or SOD2 deficiency. Therefore, this finding isn’t in good harmony with the title given to the article, because no evident cross-talk exists between p53 and SOD metabolism.

However, the manuscript is interesting because faced with a proper strategy and therefore could merit publication on Int. J. Mol. Sci. if the following concerns are carefully addressed.

-   Modify the title of the manuscript with a less ambitious sentence.

Response 1: As suggested, we have revised the title to, “Pathological relationship between intracellular superoxide metabolism and p53 signaling in mice” (page 1, line 2-3).

-   Introduction. The Authors did not discuss the properties of another important SOD, such as the extracellular enzyme (SOD3) exerting a key role in the redox control of extracellular space. In particular, it is known that SOD3 deficiency is involved in several pathologies, including cardiovascular diseases (Eur. J. Histochem 2014, 58:2383; Sci. Rep. 2020 10, 210). Therefore, this item, although not considered in the experimental strategy, should be at least discussed.

Response 2: As suggested, we have added the description, “Recently, Guo et al. reported that a loss-of-function mutation in extracellular SOD (SOD3) induced chronic kidney disease accompanied by systolic hypertension and cardiac hypertrophy in a Dahl/Salt-sensitive strain of rats [63]. In addition, SOD3 KO mice also showed hypoxia-induced pulmonary vascular disease [64, 65]. These reports suggest that not only intracellular but also extracellular ROS affect cardiac hypertrophy and cardiovascular diseases.” (page 7, line 240-245) to the discussion section and included three important references (No. 63-65) in the revised manuscript.

-   Fig. 1. In panel B the statistical significance (p < 0.01) assigned to p53–/–isn’t believable with the presented error bars.

Response 3: As suggested, we have added the description, “The relative BrdU incorporate values were calculated by a triplicate analysis.” (page 8, line 286-287) to the materials and methods section in the revised manuscript. We have also attached a data analysis file to our comment letter.

-   Fig. 1. In panel E, an obvious positive control (p53–/–) is missing; please, add this information or explain the reasons for its omission.

Response 4: As suggested, we have added the description, “p53 deficiency is well known to be incapable of promoting apoptotic cell death caused by ROS in fibroblasts [41, 42]. Our data also indicated that p53 deficiency effectively suppressed apoptosis induction via DDR in Sod1-/- cells. Therefore, DKO cells can survive despite oxidative damage under normal atmospheric conditions (Figure 1)” (page 6, line 176-180) to the discussion section of the revised manuscript.

-   Results section (page 3, lines 100-102). The statement “Unexpectedly, p53 loss did not induce SOD1-deficient phenotypes, such as body weight reduction, muscle atrophy, and liver weight gain, …” is unclear. Indeed, the data presented in Fig. 2A-C only show a minimum and not significant difference between histograms of WT and Sod1–/–; therefore, the comment related to the missing induction due to the loss of p53 (histograms of p53–/–) seems not appropriate. Only, the comment related to the skin thickness seems to be appropriate.

Response 5: As suggested, we have revised the description to, “Unexpectedly, p53 loss did not affect the body weight, muscle weight, liver weight, or skin thickness in Sod1-/- mice (Figure A-D). However, DKO mice showed exacerbated splenomegaly accompanied by a tendency toward a reduction in the red blood cell count compared with Sod1-/- mice (Figure 2E-F).” (page 3, line 102-105) in the results section of the revised manuscript.

-   Fig. 2. Please revise the indication for panels E and F, because there is an apparent inversion of letters in Figure legend.

Response 6: As suggested, we have revised the description to, “Figure 2. p53 loss had no marked effect on SOD1-decifient phenotypes in mice. (A) The body weight of each mouse (at four months of age). The genotypes for each mouse were as follows: 1. Wild type (n = 10), 2. Sod1-/- (n = 7), 3. p53-/- (n = 5), 4. Sod1-/-, p53-/- (DKO, n = 5), 5. Sod1-/-, p53+/- (n = 10), 6. Sod1+/-, p53-/- (n = 9). (B) The ratio of muscle weight corrected by body weight. (C) The ratio of liver weight corrected by body weight. (D) Hematoxylin and eosin staining, the thickness of the back skin, and the skin thickness of each mouse. (E) The ratio of the spleen weight corrected by the body weight. (F) The number of red blood cells in each mouse. Statistical analyses were performed using a two-way analysis of variance. The error bars indicate the standard deviation. *p < 0.05. **p < 0.01 vs. WT. #p < 0.05 vs. Sod1-/-. The scale bar represents 100 μm.” (page 4, line 111-119) in the results section of the revised manuscript.

-   Results section (page 5, paragraph 2.4). Please, add the meaning for genotypes p53f/fand Sod2f/fin the text.

Response 7: As suggested, we have added the description, “WT mice, including Sod2f/f and p53f/f mice,” (page 5, line 148) to the discussion section of the revised manuscript.

-   Fig. 4. In panel A, lines related to Sod2f/fand p53f/fare missing.

Response 8: As suggested, we have revised Fig. 4. In panel A (page 5, line 153).

-   Discussion. This part of the manuscript seems extremely long. This Reviewer suggests a significant shortage to allow a better focusing on the objectives reached with the research.

Response 9: As suggested, we have deleted substantial parts of the discussion section.

Reviewer 2 Report

Review is attached.

Author Response

Reviewer 2

The paper from Watanabe et al. describes the phenotype of SOD1/p53 double mutant and SOD2-hart-specific /p53 mutant mice. They found that these model animals can be successfully used in ageing studies.

I have to confess that this is the best paper in this year what I have evaluated as a reviewer. The goals are well defined and the data sets gained from the experiments were solid.

Suggestions:

Include a paragraph describing the significance of SOD1 and SOD2 in human cases. Please, include some sentences dealing with the proven data about SOD1 and SOD2-driven human diseases.

The authors mention other studies where SOD mutant mice were used, and compare the results with their mutant mice. Please, check the strain similarities or differences between the SOD mutant mice; for example, when you mention SOD mutant mice from reference 32 and 11.

Response 10: As suggested, we have added the description, “Since SOD expression and activity are significantly decreased in aged osteoporotic, end-stage osteoarthritic, and Alzheimer’s disease individuals [4-6], redox imbalance caused by SOD decline is considered an important mechanism underlying the induction of age-related pathological changes.” (page 1, line 32-35) to the introduction section of the revised manuscript.

Several SOD1 KO mice have been generated and analyzed (Reaume et al., Nat. Genet. 1996;13(1):43–47.; Huang et al., Arch. Biochem. Biophys. 1997;344(2):424–432.; Matzuk et al. Endocrinology. 1998 Sep;139(9):4008-11). All of these mice have shown age-related phenotypes, such as a shortened lifespan, muscle atrophy, hearing loss, skin thinning, and cataract formation. We therefore consider the SOD1-deficient phenotypes not to be different strains from Sod1-/-mice. Two strains of Sod2 flox mice were generated (Ikegami et al. Biochem Biophys Res Commun. 2002 Aug 23;296(3):729-36.; Strassburger et al. Free Radic Biol Med. 2005 Jun 1;38(11):1458-70.). Because whole-body knockout of SOD2 using both Sod2 flox mice showed neonatal lethality, the SOD2-deficient phenotypes were extremely similar to those of flox mice.

Major modifiable text parts:

1., Include the TYPE of cell used in certain studies.

For example: in line 39 and in line 171: “Sod1-/ - cells”. Is this mean that these cells are fibroblasts?

Response 11: As suggested, we have clarified that these are “fibroblasts” (page 1, line 42, page 2, line 81-82, page 6, line 174) throughout the revised manuscript.

2., Add a sentence to the last paragraph of the Introduction describing the novel SOD mutant mice genotype.

Response 12: As suggested, we have added the description, “…we generated two types of double-knockout (DKO) mice: Sod1 and p53 DKO mice as well as heart-specific Sod2 and p53-deficient mice.” (page 2, line 65-55) to the introduction section of the revised manuscript.

3., Correct the text of the legend of Figure 1: Instead of “each fibroblast” please use “each type of fibroblasts”.

In the legend of Figure 1. and Figure 2: Include that how many mice were used in the experiments.

In Figure 3: The Fig.3/A is not informative. Need a wild type control and the description of the original organ of the tumour. I suggest to delete this Fig.3/A.

Response 13: As suggested, we have revised the text to “each type of fibroblast” (page 3, line 94 and 96) and added the text, “three independent experiments (n=3)” (page 3, line 98-99) to the results section. In addition, we revised the Figure 3A picture and legend as follows: “Representative DKO mice accompanied by a tumor.” (page 4, line 135-137).

  1. Explain better the 2.3. part:

-It is not clear if the tumours in the mice developed spontaneous or it was induced.

-Instead of using “tumor progression” (in line 127 and 128) better to use “tumour initiation” or “tumour development”

-Include a section in Materials and Methods describing the “Tumour development assessment”

Response 14: As suggested, we have added the text, “spontaneous” (page 4, line 125-126) and revised the text to “development” (page 4, line 133) in the results section of the revised manuscript.

  1. Add a table in the 4.1 section with the sequences of all the used primers for genotyping.

Response 15: As suggested, we have added Table S1 and the text, “Primer sequences are given in Table S1.” (page 7, line 264-265) and “Supplementary Materials: The following are available online at www.mdpi.com/xxx/s1: Table S1: List of primers used for genotyping.” (page 8, line 301-302) to the revised manuscript.

Minor modifiable text parts:

Correct the word “indistinguishable” in line 106. and 146. Instead of “indistinguishable” please use “highly similar”.

in line 149: instead of “not strongly involved p53”, use “ not influenced by the loss of p53 molecule”.

in line 152 (Figure 4 legend): instead of “largely uninfluenced by” use “mainly unaffected by”.

in line 178.: instead of “contributed little” use “mainly not influence”

in line 192-194: reformat the following two sentences starting with: “Since SOD1 activity…. …. in cytoplasm.”

in line 202-203: reformat the following sentence starting with “Thus, Sod1-deficinet phenotypes….”

Response 16: As suggested, we have revised the text to read “extremely similar” (page 3, line 107), “with strong similarity” (page 5, line 147-148), “not influenced by the loss of the p53 molecule.” (page 5, line 151-152), “largely unaffected” (page 5, line 154), “did not strongly influence” (page 6, line 183), “Since SOD1 enzyme includes copper and zinc ions, SOD1 also acts as a chelator of copper and zinc ions. Sod1 deficiency might induce an increase in free copper and zinc ions in cytoplasm.” (page 6, line 197-198), and “In addition to the loss of antioxidant activity, the loss of the metal chelating ability and transcriptional function might markedly affect Sod1-deficient phenotypes.” (page 6, line 207-208) in the revised manuscript.

Round 2

Reviewer 1 Report

Most of the concerns raised by this Reviewer were properly addressed in the revised version of the manuscript. However, there is still one point that deserves a further revision by the Authors, because the modified test in Results section (page 3, lines 101-109) remains unclear. In particular, the statement “p53 loss did not affect the body weight, muscle weight, liver weight, or skin thickness in Sod1–/–mice” is misleading, because it does not properly describe the experimental data presented in Fig. 2. Indeed, the parameters body weight and liver weight are essentially unaffected in Sod1–/–, p53–/–, or in DKO mice; the muscle weight is unaffected in Sod1–/–or p53–/–, but is significantly reduced in DKO; the skin thickness is reduced in Sod1–/–, or DKO, but is unaffected in p53–/–. Therefore, a better description of these data is recommended in order to avoid a possible confusion in the reader (“p53 loss … in Sod1–/–mice” could suggest the DKO mice). On the other hand, the following statements seem to be appropriate.

After a careful revision of this item, the manuscript could merit publication on Int. J. Mol. Sci.

Author Response

Reviewer 1

Most of the concerns raised by this Reviewer were properly addressed in the revised version of the manuscript. However, there is still one point that deserves a further revision by the Authors, because the modified test in Results section (page 3, lines 101-109) remains unclear. In particular, the statement “p53 loss did not affect the body weight, muscle weight, liver weight, or skin thickness in Sod1–/–mice” is misleading, because it does not properly describe the experimental data presented in Fig. 2. Indeed, the parameters body weight and liver weight are essentially unaffected in Sod1–/–, p53–/–, or in DKO mice; the muscle weight is unaffected in Sod1–/–or p53–/–, but is significantly reduced in DKO; the skin thickness is reduced in Sod1–/–, or DKO, but is unaffected in p53–/–. Therefore, a better description of these data is recommended in order to avoid a possible confusion in the reader (“p53 loss … in Sod1–/–mice” could suggest the DKO mice). On the other hand, the following statements seem to be appropriate.

After a careful revision of this item, the manuscript could merit publication on Int. J. Mol. Sci.

Response: As suggested, we have revised the description to, “To evaluate the effect of p53 deficiency in Sod1-/- mice, we expanded the intercrossing and analyzed organ phenotypes of DKO mice. Sod1-/- mice revealed body weight reduction, muscle atrophy, and liver weight gain [1], but these were no significant differences compared with wild-type (WT) in this analysis (Figure 2A-C). SOD1 loss significantly induced skin thinning and decrease of red blood cell number but not splenomegaly (Figure 2D-F). On the other hand, p53-/- mice showed no significant differences in all parameters (Figure 2A-F). DKO mice showed significant reductions of muscle weight, skin thickness, and red blood cell number (Figure 2B, D, and F). Interestingly, DKO mice also exhibited exacerbation of splenomegaly compared with Sod1-/- mice (Figure 2E). Importantly, p53 haploinsufficiency also failed to improve Sod1-/- phenotypes (Figure 2A-F). Furthermore, Sod1+/-, p53-/- mice were extremely similar to WT and p53-/- mice (Figure 2A-F). These data indicate that p53 insufficiency did not seriously influence the organ phenotypes of Sod1-/- mice.(page 3, line 101-113) in the results section of the revised manuscript.
